# SecIoTComm: An Actor-Based Model and Framework for Secure IoT Communication

**DOI:** 10.3390/s22197313

**Published:** 2022-09-27

**Authors:** Kelechi Eze, Ahmed Abdelmoamen Ahmed, Cajetan Akujuobi

**Affiliations:** 1Center of Excellence for Communication Systems Technology Research (CECSTR) and SECURE Cybersecurity Center of Excellence, Roy G. Perry College of Engineering, Prairie View A&M University, Prairie View, TX 77446, USA; 2Department of Computer Science, Roy G. Perry College of Engineering, Prairie View A&M University, Prairie View, TX 77446, USA

**Keywords:** security, actors, Internet of Things, distributed computation, communication

## Abstract

Internet of Things (IoT) ecosystems are becoming increasingly ubiquitous and heterogeneous, adding extra layers of complexity to secure communication and resource allocation. IoT computing resources are often located at the network edge and distributed across many heterogeneous sensors, actuators, and controller devices. This makes it challenging to provide the proper security mechanisms to IoT ecosystems in terms of manageability and maintainability. In an IoT ecosystem, computational resources are naturally distributed and shareable among their constituency, which creates an opportunity to distribute heavy tasks to them. However, resource allocation in IoT requires secure and complex communication and coordination mechanisms, which existing ones do not adequately support. In this paper, we present Secure Actor-based Model for IoT Communication (SecIoTComm), a model for representing secure IoT communication. SecIoTComm aims to represent secure IoT communication properties and design and implement novel mechanisms to improve their programmability and performance. SecIoTComm separates the communication and computation concerns, achieving design modularity in building IoT ecosystems. First, this paper presents the syntax and operational semantics of SecIoTComm. Then, we present an IoT framework implementing the key concepts of the model. Finally, we evaluate the developed framework using various performance and scalability metrics.

## 1. Introduction

The Internet of Things (IoT) creates opportunities for the provision of data-driven services in many shapes and forms [1,2,3]. For instance, consider a remote patient monitoring IoT system serving physicians who need to monitor their patients’ health conditions. Another example in the agricultural sector is a livestock-monitoring IoT system that utilizes in situ sensors mounted on a livestock farm to collect data regarding cattle’s location, well-being, and health. This information would help farmers identify diseased animals so that they can be separated from the herd, thereby preventing the spread of diseases [4].

The IoT ecosystem is an engineered system that integrates computational algorithms with physical sensing components and processes. The computational algorithms coordinate and communicate with sensors that monitor cyber and physical indicators, along with actuators that modify the cyber and physical environment [5]. Such intelligent systems use sensors to connect all distributed intelligence in the environment to gain a more in-depth knowledge of the environment, enabling more accurate actions and tasks. Such an IoT ecosystem can scale to billions of end devices (i.e., sensors and actuators) connected to gateways, which act as the aggregation points for a group of sensors and actuators to coordinate the connectivity of these devices to each other and an external network [6].

The Infrastructure behind these IoT systems comprises the sensing infrastructure (i.e., sensors and actuators) and cloud infrastructure (i.e., computing and storage servers) integrated with user infrastructure (i.e., computers, mobile phones, etc.) to form a heterogeneous IoT ecosystem. The IoT gateway needs to authenticate all IoT devices participating in the sensing process by executing various cryptographic computations on the edge of the network. However, IoT devices usually have limited resources, which complicates secure communication.

In addition to simple synchronous or asynchronous communication, IoT systems often involve different types of group communications, with both multiple recipients as well as multiple senders [7]. For example, consider several in situ sensors autonomously sending their feeds to many servers to be used in aggregate form as the basis for an IoT system. What is required for supporting this are high-level communication primitives, which do not mix functional concerns of services with complex communication concerns.

In this paper, we are interested in separating secure communication concerns from IoT computations. This is important for two reasons: (i) IoT computation can apply different security mechanisms to communication at the creation time, and (ii) communication can evolve over time following its own separate logic. We propose SecIoTComm, Secure Actor-based Model for IoT Communication, a model and framework for representing and implementing secure IoT communication, respectively.

SecIoTComm extends the Actor model of concurrency [8]. Actor computations are made up of primitive agents called actors, each of which encapsulates an object with the thread of control executing it. Actors have globally unique names. An actor can create new actors and can communicate with other actors using asynchronous messages. The model’s properties of encapsulation, fair scheduling, location transparency, and locality of reference mean that languages implementing actors abstract over execution details such as processes’ physical location, scheduling, name resolution, and buffering of communication.

The contributions of this paper are fourfold. First, we present the syntax and operational semantics of SecIoTComm. The syntax defines the SecIoTComm configuration, and the semantic represents the meaning of secure IoT operations, including communication, computations, and security operations. Second, we show how complex communications can be built by composing simpler ones and give examples to demonstrate the concept of communication composition. SecIoTComm can be used to understand and represent the secure communication and coordination requirements of a wider class of sensor-based systems. We also present the syntax and operational semantics of these composition rules with examples. Third, we present an actor-based framework implementing the SecIoTComm model using Scala/Akka programming language [9]. Finally, we evaluated the developed framework using various experimental experiments.

The rest of the paper is organized as follows. We review related work in Section 2; Section 3 and Section 4 present the definition and operational semantics of SecIoTComm, respectively. Section 5 describes a set of composition rules through which more complex communications are constructed from simpler ones. Section 6 presents the framework implementation of SecIoTComm. Section 7 presents our experimental results showing processing overhead and scalability of the framework. Section 8 concludes this paper.

## 2. Related Work

Although there are some technologies focused on IoT systems [2,6,10,11], there is relatively limited foundational work. This is in part because of the lack of precise understanding, specification, and analysis of such systems, and consequently, there is limited platform support for programming them.

The programming required for offering a new sensor-based system can be significant if started from scratch. However, there is an opportunity created by the similarity in the patterns of communication required for such IoT systems, where contextual data offered by a number of contributors become the basis for the system. This pattern of communication was originally defined in [12] as multi-origin communication. Multi-origin coordination mechanisms can be provided on a platform over which such class of services could be implemented relatively easily.

Ahmed et al. interpreted and implemented these mechanisms for the domain of crowd-sourced services [13,14] by implementing CSSWare, a middleware which provides a set of domain-specific mechanisms to support initiating and managing services. Having CSSWare as a platform, all that a service designer needs to do to launch a new service is to identify a constituency of potential contributors and to provide a few lines of service-specific code for specifying the nature of contributions and for aggregating them when they arrive.

In [15], Agha presented a coordination model for large-scale distributed actor systems. The developed model allows actors to adapt the underlying communication mechanisms to support such systems. The author presented a scoped semantics for the proposed coordination model, named synchronization model, based on various declarative synchronization constraints. Similarly, in [16], Frølund presented another coordination approach, allowing the coordination of distributed application components in the form of abstract and reusable coordination constraints. Christian J. et al. [17] presented ActorSpace, a new programming paradigm to provide a communication model in the form of actor context for pattern matching on certain actor attributes to enable communication to certain enabled groups defined by the patterns.

Interactors [18], which is the most relevant work to our model, is a model for representing complex communication in distributed systems. Interactors aims to separate the communication concerns from functional concerns of such systems, whereas communication is represented as self-driven and can dynamically change as per communication needs. The authors presented the syntax and operational semantics of multicast, which is a richer form of communication pattern, as a feasible solution for implementing complex types of communications in actor-based systems. Prokopec et al. [19] proposed Reactive, a model to simplify the composition of communication protocols using first-class channels and event streams. Reactive aims to overcome the obstacles in composing classic actors.

HL et al. [20] presented an actor-based implementation of the actor model in the IoT domain, emphasizing its event-driven and reactive features. Another work [21] evaluated the suitability of the actor model for IoT by comparing its performance on blockchain in accordance with best practice implementation based on key IoT performance requirements.

In summary, most of the existing works in the area of secure IoT communication [2,6,20,21,22] mix between the computation and communication concerns without addressing the security aspects for limited-resource IoT devices. In comparison, SecIoTComm separates the secure communication and actor computations aspects by composing simple communication to realize complex communication operations via different composition rules. To the best of our knowledge, no actor-based framework for secure IoT communication exists in the literature that supports communication composition. We also evaluated our framework with several using various performance and scalability metrics.

## 3. Secure Actor-Based Model for IoT Communication

We model the physical IoT devices as Actor Systems (AS) and software components hosted on a given AS represented by individual actors. Computations are carried out on a given AS using a set of actors represented as ports, which can be one of two types: input or output. Ports interact with other ports by sensing and receiving asynchronous actor messages. Complex communication can be created by composing simple forms of communication using defined composition rules. Two IoT devices can establish a secure communication by exchanging their ports’ names (addresses). Ports are responsible for applying the required security mechanisms such as authentication and encryption of actor messages. Security mechanisms are often predefined in a security configuration in the AS as security primitives. Table 1 summarizes the mathematical symbols used in SecIoTComm.

Figure 1 illustrates an example scenario for the interaction between IoT devices. A rectangle represents an AS’s boundary, representing a single IoT device. Each device has a set of ports using which it could communicate with other devices in the IoT ecosystem. The figure also illustrates the components of IoT devices in our model. White circles are input ports, black circles are output ports, and the lines with arrows represent message flows. Actor Systems ($AS_{1}$ … $AS_{3}$) are IoT gateways collecting sensor data from environmental via various heterogeneous sensors. $AS_{4}$ can be a local data logger on the network edge, used for aggregating sensor feeds data for local data processing before sending the aggregate to the cloud. $AS_{5}$ is a user device that consumes the services provided by the local server on the network edge.

SecIoTComm configuration helps in studying the properties of secure IoT systems in depth at runtime. We formally define SecIoTComm configuration *k*, which is a snapshot of the actor system from an idealized observer, capturing an instantaneous global state of the system as a 6-tuple as follows:(1)$${\langle A|C|S|M|\alpha |\mu |\rangle }_{\chi }$$
where *A* is a finite set of computations; *C* is a finite set of communication; *S* is a finite set of security configuration; *M* is a finite set of messages *m* in the communication layer; α is a mapping of actor addresses to their behaviors (actor map); μ represents transient, buffered, or floating message in the system; and χ are the external actors.

A message *m* has two parts: message content *v* and recipient *a*, represented as a⇐v. *A*, *C*, and χ are represented in our model as sets of actor addresses, whereas A=Dom(a) and ρ⊂A. The following two constrains must hold:(2)ρ⊆AandA∩χ=Ø
(3)ifa∈A,thenFV(α(a))⊆A∪χ
where FV is a free variable of α(a). The receptionist of *k* is $\overset{n}{\underset{i=1}{⋃}}ip_{i}$ where $ip_{i}$ is the ith input port of *k* and *n* is the number of such ports in *k*.

### 3.1. Computation Definition

We define the IoT computation, *A*, as a reduction process with a finite set of labeled transitions, *T*, represented as a triple-tuple:(4)(I:T:O)
where *I* is a set of actors in their initial computation states modeled as input ports (inports) or ip, and *O* is a set of actors in their final computation states modeled as output ports (outports) or op that are ready to create a secure communication with other actors i∈I hosted in other actor systems.

Communication is initiated from the outports of one AS to the inports of another one. inports and outports represent the computing actors in the communicating ASs. For simplicity, we refer to any computing actor in our model as computation; hence, ip ∈ *A* ∧ op ∈ *A*.

### 3.2. Communication Definition

Communication, $C_{c}$, is represented by a standalone channel created by a computation to enable secure communication among IoT devices. We model IoT communications as:(5)(C:J)
where *C* is a finite set of communications modeled as actors, and *J* is a mapping of communication to the computation of the form (c,ip,op), where *c* is the actual communication, ip is the target of the communication, and op is the origin of the communication. Note that the target is the same as inport, (ip), of the recipient actor system, and op is the outport of the source actor system.

### 3.3. Security Definition

Security, *S*, is defined as a set of security primitives accessible by the interacting actors in a given IoT ecosystem. It is used to secure its computations and communications. An instantaneous snapshot of all security mechanisms in the IoT system is called security configuration, which is represented as:(6)(G:X)
where *G* is the set of security primitives and *X* is the mapping of security primitives to computation. Each mapping of security primitive to computations is of the form $\left(a,k_{s}\right)$, where *a* is the computing actor that applies the security primitive $k_{s}$ ∈ *G* to communications.

## 4. SecIoTComm Operational Semantics

The SecIoTComm configuration evolves due to state changes of the hosted actors, which leads to a transition in configurations. The actor state can be one of the following:*Idle-uninitialized*, ${\left(a^{\prime }\right)}_{a}$ a newly created actor $a^{\prime }$ by an actor *a* but uninitialized.*Idle-initialized*, ${\left(a^{\prime },b\right)}_{a}$ a newly created actor $a^{\prime }$ by an actor *a* initialized to a behavior *b*.*Busy*, ${\left(a^{\prime },e\right)}_{a}$ actor $a^{\prime }$ is busy evaluating an expression *e*.

An actor transition can either be one of the two categories below:*Internal transitions or (r−x) receive/execute transitions:* receive transition consume a message where execute transition may send a message, create a new uninitialized actor, or initialized an already created actor.*External transitions or (i−o) input/output transitions:* input transition involves arrival of a message from a source external to the configuration where output configuration involves passing a message to a destination external to the configuration.

The transition relation on an actor configuration determines the future configuration. A labeled transition relation ↦ on configuration defines a set of computations of an actor configuration. Computations are reductions of expressions. Hence, transitions are a reduction process. The same rules apply to transitions and reductions, as they are synonymous.

We represent an actor expression as *e*, which can be a value expression or non-value expression e=v. A value expression is an expression, *v*, in its reduced form; hence, a non-value expression is decomposed into a reduction context *R* filled with redex (reducible expression), that is:(7)(e=R[r])
where *R* is the reduction context, and *r* is the redex We represent a transition rule by λ↦Z, where
(8)Z=Dom(α)∪{a}∪χ

### 4.1. Actor System Initialization

The Actor System creates a new actor actor and returns the actor name, as described by this transition rule: (9)$${\langle A|C|S|M|\alpha ,\left[R\left[create\_a\left(\right)\right]\right]}_{as}{|\mu \rangle }_{\chi }\mapsto \langle A|C|S|M|\alpha ,{\left[R\left[a^{{}^{\prime }}\right]\right]}_{as},{\left(a^{\prime }\right)}_{a}{|\mu \rangle }_{\chi }$$
where create_a() is a redex for actor creation, which creates the top guardian actor of the actor system.

The Actor System then initializes the behaviors of the freshly created actors with the name *a* to *b* and returns its behavior, as described by the following transition rule: (10)$${\langle A|C|S|M|\alpha ,\left[R\left[initialize\_a\left(a,b\right)\right]\right]}_{as},{\left(\right)}_{a}{|\mu \rangle }_{\chi }\mapsto {\langle A|C|S|M|\alpha ,\left[R\left[nil\right]\right]}_{as},{\left(b\right)}_{a}{|\mu \rangle }_{\chi }$$

### 4.2. Receiving a Message through ip ∈ $A_{c}$

A computation, ip, that is triggered when an actor receives a message *m* of the form v⇒a can perform one of the following actions by applying the received message to its current behavior:(i)Perform a multi-step computation;(ii)Create a communication;(iii)Store the result of a computation.

Formally, a message receipt by any actor can be described by the following transition rule: (11)$${\langle A|C|S|M,m|\alpha ,\left(b\right)}_{ip}{|\mu \rangle }_{\chi }\mapsto \langle A|C|S|M|\alpha ,{[app{\left(b,m\right]}_{ip}|M_{a}\rangle }_{\chi }$$

Below, we described the transition rules of the two cases of message receipt and their behavior:(i)*Message receipt by a computation* Message can be received on i∈I of $C_{a}$, as follows:
(12)$$\langle A,\left(I,{\left(b\right)}_{i}:T:O\right){|C|S|M|\alpha |\mu ,m\rangle }_{\chi }\mapsto \langle A,\left(I,{\left[app\left(b,m\right)\right]}_{i}:T:O\right){|C|S|M|\alpha |\mu \rangle }_{\chi }$$Message can be received on o∈I of $C_{a}$, as follows:
(13)$$\langle A,\left(I:T:O,{\left(b\right)}_{o}\right){|C|S|M,m|\alpha |\mu \rangle }_{\chi }\mapsto \langle A,\left(I:T:O,{\left[app\left(b,m\right)\right]}_{o}\right){|C|S|M|\alpha |\mu \rangle }_{\chi }$$(ii)*Message receipt by a communication*(14)$$\langle A,\left(I:T:O\right)|C,\left(C,{\left(b\right)}_{c}:J\right){|S|M|\alpha |M,m\rangle }_{\chi }\mapsto \langle A,\left(I:T:O\right)|C,\left[app\left(b,m\right)\right]{|S|M|\alpha |\mu \rangle }_{\chi }$$

### 4.3. A Computation Creates a Communication

A computation creates a communication as an independent entity with a separate communication logic to carry out its communication logic, as follows:(15)$$\langle C_{a},\left(I,{\left[R\left[new\left(b\right)\right]\right]}_{i}:T:O\right)|C_{c}{,\left(C:J\right)|\alpha |M\rangle }_{\chi }\mapsto \langle C_{a},(I,\left[R{\left[c\right]}_{i}:T:O\right)|C_{c},\left(C,{\left(b\right)}_{c}:J\right){|\alpha |M\rangle }_{\chi }$$

### 4.4. A Communication Receives a Message

A communication can receive a message from another communication or computation. In both cases, the message is applied to its current behavior, as follows: (16)$$\langle C_{a},\left(I:T:O\right)|C_{c},\left(C,{\left(b\right)}_{c}:J\right){|\alpha |M,m\rangle }_{\chi }\mapsto \langle C_{a},\left(I:T:O\right)|C_{c},\left(C,{\left[app\left(b,m\right)\right]}_{c}:J\right){|\alpha |M\rangle }_{\chi }$$

### 4.5. A Communication Sends a Message

A communication can send a message to another communication or computation. The first case models actor routing, while the second case models a response or an acknowledgment. The latter case generates a message that is sent in the communication channel towards its destination, as described below: (17)$$\langle A|C,\left(C,{\left[R\left[send\left(m\right)\right]\right]}_{c}:J\right){|S|M|\alpha |\mu \rangle }_{\chi }\mapsto \langle A|C,\left(C,{\left[R\left[nil\right]\right]}_{c}:J\right)\left|S|M|\alpha \right|\mu^{\prime }\rangle_{\chi }$$

### 4.6. Secure Communication

Secure communication is initiated between two or more Actor Systems such that $k_{s}\in G\neq null$, where $k_{s}$ is the security configuration for this communication. The following equations represent the process of initiating a secure communication between Actor Systems:(18)$${\langle A,\left(I:T:O\right)|C|S|M|\alpha ,\left[R\left[send\left(m\right)\right]\right]}_{c}|\mu \rangle \mapsto \langle A,\left(I:T:O\right)|C|M^{\prime }{|S|\alpha ,\left[R\left[nil\right]\right]}_{c}|\mu^{\prime }\rangle$$
(19)$${\langle A,\left(I:T:O\right)|C|S|M|\alpha ,\left(b\right)}_{c}\left|\mu ,c\leftarrow conf\rangle \mapsto \langle A,\left(I:T:O\right)|C\right|M^{\prime }{|S|\alpha ,\left[app\left(b,ip\right)\right]}_{c}|\mu^{\prime }\rangle$$
where *m* is the message that initiates the security configuration and conf is the created security configuration.

Secure communication is established between Actor Systems when a computation actor hosted in the sender Actor System sends a message *m* to a communication *c* to retrieve a security configuration conf. This process is formally represented as:(20)$$\langle A,\left(I,{\left[R\left[send\left(m\right)\right]\right]}_{ip}:T:O\right)|C|S|M|\alpha |\mu \rangle \mapsto \langle A,\left(I{\left[R\left[nil\right]\right]}_{ip}:T:O\right)\left|C\right|M^{\prime }\left|S|\alpha \right|\mu^{\prime }\rangle$$
where m=h←conf.

Once the security configuration is retrieved from the device memory, it is immediately applied to the existing communication, as follows:(21)$$\langle A,\left(I:T:O\right)|C,\left(H,{\left(b\right)}_{h}:J\right)|S|M|\alpha |\mu \rangle \mapsto \langle A,\left(I:T:O\right)|C,\left(H,\left[app\left(b,m\right)\right]\right)|M^{\prime }\left|S|\alpha \right|\mu^{\prime }\rangle$$

## 5. Composition Semantics and Communication Patterns in SecIoTComm

We define communication compositionally. In other words, simpler communication can be composed to create more complex ones by applying a set of composition rules. A communication connects to other communications using input and output ports. Messages can be received by a communication at its input ports and can be sent out to other communications from its output ports. An external observer can only see the input ports and the output ports of the final composed communication.

The composition of two communications $k_{1}$ and $k_{2}$ is represented as $k_{1}\cup_{\|}k_{2}$. Two communications $k_{i}=\langle A_{i}|C_{i}|S_{i}|M_{i}|\alpha_{i}{|\mu_{i}\rangle }_{\chi_{i}}$, 0<i≤2 are composable if $Dom\left(\alpha_{1}\right)\cap Dom\left(\alpha_{2}\right)=Ø$, $\chi_{1}\cap Dom\left(\alpha_{2}\right)\subseteq \rho_{2}$, and $\chi_{2}\cap Dom\left(\alpha_{1}\right)\subseteq \rho_{1}$, where $\rho_{1}\subset C_{a_{1}}$ and $\rho_{2}\subset C_{a_{2}}$.

The remainder of this section presents the semantics of communication composition. We abstract away from the representation of communication internal messages because we are only concerned about the coming together of communications.

### 5.1. Input Convergence Composition

Input convergence composition can be used to implement a one-to-many communication pattern in SecIoTComm. We define one-to-many communication as a type of communication involving a single actor from an actor system $AS_{1}$ communicating to *n* number of actors in another system $AS_{2}$.

As shown in Figure 2, the input convergence composition takes two parameters as input: a target set *D* of computing actors and a behavior *b* informing the source computing actor about the number of communication to create. The newly created communication actors make the different targets of the source computing actor their targets and inherit the original behavior of the source computing actor. Then, the source computing actor changes its behavior to the new provided behavior, which forwards its messages to different communication actors who already know its targets. This can be represented formally as follows:(22)$$\begin{array}{c} \cup_{{\left|\right|}_{i=1}}^{\;\;n}D,b\langle \left(I_{i}:T_{i}:O_{i}\right)|\left(C_{i}:J_{i}\right)\left|\alpha_{i}\right|M_{i}\rangle_{\chi_{i}}\Rightarrow \\ \langle \left(\left(\overset{n}{\underset{i=1}{⋃}}I_{i}-S\right)\cup \overset{m}{\underset{j=1}{⋃}}{\left(b_{j},S\right)}_{c_{a}}:\overset{n}{\underset{i=1}{⋃}}T_{i}:\overset{n}{\underset{i=0}{⋃}}O_{i}\right)|(\left(\overset{n}{\underset{i=1}{⋃}}C_{i}\cup C_{new}\right):\\ \overset{n}{\underset{i=0}{⋃}}J_{i}\cup \left(c,S,D\right))|⋃\left\{i=0\;\;n\right\}\;S|\overset{N}{\underset{i=1}{⋃}}\alpha_{i}|\overset{n}{\underset{i=1}{⋃}}M_{i}\rangle_{\overset{N}{\underset{i=1}{⋃}}\chi {}_{i}-\overset{n}{\underset{i=1}{⋃}}\rho_{i}} \end{array}$$
where *D* is the set of recipients (targets) to be composed, *b* is the new behavior that is passed as a parameter to the composition, and *S* is a set of all sources of communication. Note that *D* and *S* are computing actors and each member of *D* communicates with all members of *S*. $C_{new}$ is the set of new communications created using the create_a() redex of actor creation to carry out the communication. {i=0n} is the range of *i* values, and *S* is a security configuration. *N* is the total sum of sources and targets. The actor map of the composition is obtained by including the actor map of the new communications in the complete map of the computations in the system, which consists of sources and targets.

### 5.2. Output Convergence Composition

Output convergence composition can be used for many-to-one communication patterns [5]. We define many-to-one communication as communication involving many senders (actors) hosted in a source actor system $AS_{1}$ targeting one recipient hosted on another actor system $AS_{2}$.

As shown in Figure 3, the output convergence composition takes three parameters: a set of sources (senders) *S*, a behavior *b*, and one destination *d*. The set of sources informs the composition of the number of new communications to create, represented by a directional line with an arrow showing the direction of communication. This can be represented formally as follows:(23)$$\begin{array}{c} \cup_{{\left|\right|}_{i=1}}^{\;\;n}S,d,b\langle \left(I_{i}:T_{i}:O_{i}\right)|\left(C_{i}:J_{i}\right)\left|\alpha_{i}\right|M_{i}\rangle_{\chi_{i}}\Rightarrow \\ \langle (\overset{n}{\underset{i=1}{⋃}}I_{i}:\overset{n}{\underset{i=1}{⋃}}T_{i}:\overset{n}{\underset{i=1}{⋃}}O_{i}\cup {\left(b,d\right)}_{a_{i}^{{}^{\prime }}}\;|\;\overset{m}{\underset{j=1}{⋃}}C_{j}\;|\;\overset{m}{\underset{j=1}{⋃}}J_{j}\;\\ \left|⋃\left\{i=0\;\;n\right\}\;S\right|\overset{n}{\underset{i=1}{⋃}}\alpha_{i}\cup \overset{m}{\underset{j=1}{⋃}}\alpha_{j}\cup \left(b_{a_{i}},a_{i}\right)\;|\overset{m}{\underset{j=1}{⋃}}M_{j}\rangle_{\overset{m}{\underset{j=1}{⋃}}\left(\chi {}_{i}-\rho_{i}\right)} \end{array}$$
where $C_{j}$ is a set of new communication created by invoking create_a() function. This results in creating a new target actor $a_{i}$ with behavior *b*, and *d* is the destination actor. The length of the set of mapping of computation communication is equal to the number of involved communications. The total number of mapping of actor to behavior is obtained by summing the total number of mapping of original computations, the new computation, and the new communications. *M* is the number of message channels.

### 5.3. Complementary Composition

Given two or more SecIoTComm configurations, $k_{1}$ and $k_{2}$, many-to-many communication between $k_{1}$ and $k_{2}$ involves multiple computing actors on $k_{1}$ communicating with multiple computing actor on $k_{2}$ via complex communication patterns. This can be formally represented as:(24)$$k_{1}=\langle A_{1}|C_{1}|S_{1}|M_{1}|\alpha_{1}{|\mu_{1}\rangle }_{\chi_{1}}$$
(25)$$k_{2}=\langle A_{2}|C_{2}|S_{2}|M_{2}|\alpha_{2}{|\mu_{2}\rangle }_{\chi_{2}}$$
(26)$$k_{3}=\langle A_{3}|C_{3}|S_{3}|M_{3}|\alpha_{3}{|\mu_{3}\rangle }_{\chi_{3}}$$

As shown in Figure 4, the complimentary composition takes one parameter: a group set *G* in the form of $\overset{m}{\underset{j=1}{⋃}}\left({ip}_{j},{op}_{j},b_{j}\right)$. Below is the formal representation of the complimentary composition of *n* communications:(27)$$\begin{array}{c} \cup_{{\left|\right|}_{i=1}}^{\;\;n}G,b\langle \left(I_{i}:T_{i}:O_{i}\right)|\left(C_{i}:J_{i}\right)|S_{i}\left|\alpha_{i}\right|M_{i}\rangle_{\chi }\Rightarrow \\ \langle \left(\overset{n}{\underset{i=1}{⋃}}I_{i}\;\cup \;\overset{n}{\underset{i=1}{⋃}}T_{i}\;\cup \;\overset{n}{\underset{i=1}{⋃}}O_{i}\;\cup \;{\left(b,a^{{}^{\prime }}\right)}_{op}\right)\;|\;\left(\overset{m}{\underset{j=1}{⋃}}C_{j}\;\cup \;\overset{q}{\underset{l=1}{⋃}}C_{l}\right)\\ :\left(\overset{m}{\underset{j=1}{⋃}}J_{j}\;\cup \;\overset{q}{\underset{l=1}{⋃}}J_{l}\right)\;|\;|S_{i}|\left(\overset{N}{\underset{i=1}{⋃}}\alpha_{i}\;\cup \;\overset{n}{\underset{j=1}{⋃}}\alpha_{j}\;\cup \;\overset{q}{\underset{l=1}{⋃}}\alpha_{l}\;\cup \;\left(b,a^{{}^{\prime }}\right)\right)\\ \;|\;M_{1}\;\cup M_{2}\;\rangle_{\left(\chi {}_{1}\cup \chi_{2}\right)-\left(\rho_{1}\cup \rho_{2}\right)} \end{array}$$

#### Port-Binding Composition

Port binding is a general-purpose composition that merges the outputs ports of one actor system (or composition) with the inputs ports of another actor system or composition. To compose two actor systems configurations $k_{1}$ and $k_{2}$, we provide a set of triples for port-to-port binding whose entries are of the form $\left(op_{i},ip_{j},b\right)$, where $op_{i}$ is the ith output port of $k_{1}$, $ip_{j}$ is the jth input port of $k_{2}$ to be merged, and *b* is the provided behavior.

As shown in Figure 5, the port-to-port composition of *n* communication can be formally described as:(28)$$\begin{array}{c} \cup_{{\left|\right|}_{i=1}}^{\;\;n}op,ip,b\langle \left(I_{i}:T_{i}:O_{i}\right)|\left(C_{i}:J_{i}\right)\left|\alpha_{i}\right|M_{i}\rangle_{\chi_{i}}\Rightarrow \\ \langle \left(\overset{n}{\underset{i=1}{⋃}}I_{i}\;\cup \;\overset{n}{\underset{i=1}{⋃}}T_{i}\;\cup \;\overset{n}{\underset{i=1}{⋃}}O_{i}\right)\;|\;\overset{m}{\underset{j=1}{⋃}}C_{j}\;:\;\overset{m}{\underset{j=1}{⋃}}J_{j}\\ \;|\;⋃\left\{i=0\;\;n\right\}\;S|\left(\overset{N}{\underset{i=1}{⋃}}\alpha_{i}\;\cup \;\overset{n}{\underset{j=1}{⋃}}\alpha_{j}\;\right)\;|\;M_{1}\;\cup M_{2}\;\rangle_{\left(\chi {}_{1}\cup \chi_{2}\right)-\left(\rho_{1}\cup \rho_{2}\right)} \end{array}$$
where $C_{new}=\overset{m}{\underset{j=1}{⋃}}C_{j}$ is the set of new communications created that connects the output ports of $k_{1}$ to the input ports of $k_{2}$. $J_{new}=\overset{m}{\underset{j=1}{⋃}}J_{j}$ is the associated new mapping of communications to computations. The actor map of the port binding composition is generated by adding the map of the newly created communication to the composite map of the participating actor system. The set of messages becomes the composite messages of the two composed configurations $k_{1}$ and $k_{2}$. ρ is the receptionists and equivalent to the inports (ip) of the actor system.

## 6. SecIoTComm Implementation Framework

The design of the implemented framework builds on the domain-specific mechanisms for SecIoTComm described in the previous sections. Specifically, we built a set of programming constructs implemented in a framework to support the programmability of IoT secure systems. We prototyped the distributed framework as an actor system. The distributed run-time system for the proposed framework is organized with parts executing on the IoT end-devices at the sensing side and remote servers on the platform side.

Figure 6 illustrates the system design of the SecIoTComm implementation framework. Our prototype used four Raspberry Pi devices, a set of various sensors, and a MacBook server to represent the remote server. The IoT sensors represent the sensor data sources. Each Raspberry Pi device and the MacBook server hosts an actor system, which holds the security configurations.

Each Raspberry Pi 4 device is equipped with a 1.5GHz Quad-core Cortex-A72 (ARM v8) 64-bit SoC, 8GB LPDDR4-3200 SD-RAM, 5.0 GHz IEEE 802.11ac wireless, Bluetooth 5.0, and BLE Gigabit Ethernet, running Raspbian OS. We also used one MacBook laptop with an M2 GPU (8-core) processor and 16GB of RAM. The other security components, such as IoT protocols, communications, and computations, are represented by individual actors. We used Scala version 2.1.3 with Akka version 2.6.19 running on JVM 1.8. We set the minimum and the maximum number of active threads in the pool of threads, called parallelism-min and parallelism-max, to 8 and 64, respectively. The parallelism factor is set to 8.

The SecIoTComm framework is implemented using Akka [9], an open-source toolkit for building actor-based concurrent applications using the Scala programming language. Akka can be used to create highly concurrent, distributed, and fault-tolerant applications which can span across multiple processor cores and networks. Akka provides a high-level abstraction to low-level synchronization mechanisms of multi-threaded applications by hiding them from programmers, allowing them to focus on application-specific details.

We used several routing, clustering, and networking features embodied in Akka to realize the network architecture of the SecIoTComm framework. SecIoTComm communication is established through actor messages between JVMs residing on each Actor System hosted on each Raspberry Pi. Communication among JVMs is encrypted using the developed security configuration via the mTLS (mutual Transport Layer Security) protocol.

### 6.1. Key Generation and Secure Communication

We used two Java tools, keytool and pwgen, for the generation of security keys and passwords used in the security configurations, respectively. Security keys are generated by specifying the parameters of the keys such as alias, algorithm, keysize, validity, etc. Security keys are one of two types: private and public. As shown in Figure 7, private keys are unique, self-signed, and stored in a password-protected keystore in the device’s memory. It is used for exchanging encrypted messages among ActorSystems via asymmetric encryption.

Public keys are stored in a password-protected truststore in the device’s memory and contain many public keys of other devices in the framework’s trusted list. Figure 8 shows the keystore of Actorsystem3 (as3) that contains actorsystem1 (as1) and actorsystem2 (as2) as trustee devices.

Additionally, the framework poses the following security features:*Immutable messages:* Messages are constructed using case classes in Scala, so any malicious party cannot mutate them.*Immutable State:* We used Actors’ property of immutability state as a layer of security on actor computation.

### 6.2. Class Diagram

Figure 9 illustrates the class diagram of the framework running on Raspberry Pi devices. Actor is a top-level actor class that implements a receive method for changing the actor’s behavior when receiving a new message. The Behavior class implements the security functionalities of the framework, including: (i) the retrieve(key) function, used to retrieve the associated security configuration of a public key; (ii) the encrypt (key, target) function, used to encrypt the target configuration given a private key; (iii) the delegate (m, target) function, which sends a delegation task to another actor in the ActorSystem; and record(content), used to store a security configuration. Computation class extends the Actor and Behavior classes. It implements the computational functionalities of the framework.

### 6.3. Runtime Event Diagram of the SecIoTComm Framework

Figure 10 illustrates the runtime event diagram for the SecIoTComm framework. The runtime events diagram illustrates the framework functions, communication properties, actor computations, scalability, and resource sharing among Actor Systems. The horizontal dotted lines represent the time of each event (e.g., t1, t2, t3, and t4). Events can be classified as computation, communication, or actor creation event.

As shown in the figure, the execution time of a computation depends on the length of its computational sequence. When an ip actor carries out a computation, it becomes in the busy state during which it cannot process any other messages until the current state becomes idle. The computation time, $t_{i}$, ranges from 0 to <*∞* for non-value expressions. An (ip) actor becomes (op) after the execution of a computation.

When an ActorSystem wants to communicate with another ActorSystem, the (op) actors of the former initiate the communication process by establishing a communication channel using a predefined security configuration. The communicated actor message, along with its metadata, is received by the inport of the target ActorSystem. The execution time of the message is represented by t<∞.

Communication in SecIoTComm is asynchronous as a computation op creates a finite number of communication actors to carry out multiple communication concurrently. Computation, on the other hand, is concurrent for different ports. Ports $ip{}_{1}$ and $ip{}_{2}$ belonging to the same ActorSystem can carry out computation concurrently with execution time $t_{1}$ and $t_{2}$, respectively. Distributed computation is defined by task delegation, where an ($ip_{1}$) actor sends a computational task to ($ip_{3}$) on another ActorSystem to carry out a certain computation and send back the result.

## 7. Evaluation

This section presents our experimental evaluation of the SecIoTComm framework for performance and scalability. We compared the performance of two cipher suites encryption algorithms running on the top of our framework: TLS_DHE_RSA_WITH_AES_128_GCM_SHA256 and TLS_ECDHE_RSA_WITH_AES_128_GC_SHA256. Both cipher suites are recommended by the Internet Standard security (RFC 7525) for IoT systems. Our study evaluated various key sizes from 512 bits up to 4096 bits. We observed the impact of changing the key size on memory consumption and CPU computational cycles.

### 7.1. Performance

Figure 11 shows the CPU and memory utilization of Raspberry Pi 1 in the idle JVM state when having a fresh security configuration in place. The idle JVM state of a Raspberry Pi device happens when no active functionalities are running on the device, such as communication, computation, or security-related activities. As shown in the figure, the total CPU and memory utilization after initializing the security configurations are negligible overheads at the idle JVM state.

Figure 12, Figure 13, Figure 14, Figure 15, Figure 16, Figure 17, Figure 18 and Figure 19 show the CPU and memory utilization of Raspberry Pi 1 and 2 devices when using different cipher suites with various key sizes. The spike in CPU initialization at the first 25 seconds of the experiment resulted from the overheads of initiating the TCP communication (e.g., three-way handshake process).

Figure 12 shows the CPU and memory utilization when using the SHA256 encryption standard with keysize 512 bits in Galois-Counter mode. We observed improvement in the peak CPU consumption overhead caused by TLS handshake from 170% to 160%. The handshake duration and average CPU consumption after handshake completion remain the same at approximately 23 s and 5%, respectively. The memory overhead for this configuration is about 40Mb. The memory usage remains comparably the same for Figure 12, Figure 13 and Figure 14.

Figure 13 shows the impact of doubling the keysize from 512 to 1024 bits on the performance. As shown in the figure, the CPU consumption’s peak is pushed from 160% to 165%, while the handshake duration and average memory utilization remain the same. The memory overhead for this configuration is 35 Mb.

Figure 14 shows the Raspberry Pi device in the active state of communication using Transport Layer Protocol (TLS), Diffie–Hellman Ephemeral (DHE), and Rivest Shamir Adleman algorithm (RSA) for the exchange of cipher suites, authentication, and creating and exchanging session keys. Advanced Encryption Standard AES is used for encryption and Secure Hash Algorithm 256 (SHA256) for integrity check. SecIoTComm framework uses distributed authenticated encryption, where the handshake process is distributed between IoT devices involved in communication.

As shown in Figure 14, the TLS handshake occurs in the first 23 s of the experiment. The TLS handshake increases the CPU consumption up to 170%. Such overhead drops to about 5% on average during the remaining time of the experiment. The overall memory overhead is 37 Mb. Comparing the average CPU and memory overhead illustrated in Figure 14 with the idle state displayed in Figure 12, we can conclude that the SecIoTComm framework is lightweight and suitable for resource-constrained IoT devices.

In Figure 15, the keysize is doubled again to 4096 bits, while keeping all other experiment’s parameters the same. The TLS handshake increases the peak CPU utilization to 196% within the first 30 s of the experiment. Upon completing the TLS handshake process, the average CPU utilization remained similar to Figure 14. The overall memory overhead was 40 Mb.

In Figure 16, we evaluated the performance using Elliptic Curve Diffie–Hellman Ephemeral (ECDHE) protocol for key exchange, Rivest Shamir Adleman algorithm (RSA) for message authentication, Advanced Encryption Standard in Galois-Counter mode for encryption, and Secure Hash Algorithm 256 (SHA256) hash for integrity check.

We observed that the CPU and memory utilization are similar to the Diffie–Hellman Ephemeral (DHE) key exchange scheme used in the previous three experiments (Figure 13, Figure 14 and Figure 15); however, the ECDHE variant reduces the computation cost. The peak CPU utilization during TLS handshake was measured to be 164% during the first 25 s of the experiment. After completing the TLS handshake process, the average CPU utilization was 5%, and the overall memory overhead was 30 Mb.

In Figure 17, we evaluated the performance using ECDHE, RSA, and SHA256 using a keysize of 512. The peak CPU utilization during TLS handshake was measured to be 155% during the first 25 s of the experiment. After completing the TLS handshake process, the average CPU utilization was 5%, and the overall memory overhead was 35 Mb.

In Figure 18, we evaluated the performance using ECDHE, RSA, and SHA256 using a keysize of 1024 bits. The peak CPU utilization during TLS handshake was measured to be 163% during the first 25 s of the experiment. After completing the TLS handshake process, the average CPU utilization was 5%, and the overall memory overhead was 45 Mb.

In Figure 19, we evaluated the performance using ECDHE, RSA, and SHA256 using a keysize of 4024 bits. The peak CPU utilization during the TLS handshake was measured to be 183% during the first 25 s of the experiment. After completing the TLS handshake process, the average CPU utilization was 5%, and the overall memory overhead was 50 Mb.

Table 2 and Table 3 summarize the results of the performance of the DHE and ECDHE experiments with different key sizes, respectively.

### 7.2. Scalability

We ran a set of experiments to determine the impact of changing the number of actors on computational CPU utilization at the gateway. The number of actors determines the number of sensors a gateway can support, where each actor manages that sensor’s communication, processing, and storage need.

In the first experiment, we used one actor to perform all computations. Then, we gradually doubled the number of actors in the following experiments. As shown in Figure 20, as the number of actors used in the computations increases, the average CPU consumption increases until reaching 5000 actors. As we increase the number of actors in the communication layer to 5, 10, 50, 100, 500, and 1000 simultaneous actor communications, the average CPU utilization reaches up to 5%, 6%, 10%, 15%, 21%, and 25%, respectively. This shows the scalability of a single Raspberry Pi device running the SecIoTComm framework.

At the point of using 5000 actors, we could not observe apparent differences in CPU utilization. This may be justified by the extra overheads of initiating the actors and communication delay. These results suggest that having a large number of actors is not necessary to improve the overall performance of intensive computations. Therefore, programmers must find an equilibrium between the number of actors and CPU cycles required to perform the calculations.

### 7.3. Computational Task Delegation

We implemented a fine-grained resource coordination and control mechanism through our framework’s computational task delegation feature. Computational task delegation enables IoT devices to discover and utilize the computational resources of other devices in the network. Computational task delegation is implemented using inports (ip) and outports (op) actors. Tasks messages are received through ips actors, executed, and sent out through ops actors. An op actor creates communication with a designated behavior appropriate for its communication of three types: one-to-one, one-to-many, and many-to-one communications.

Figure 21 shows the effect of delegating a computationally intensive task from a Raspberry Pi device to a remote server (i.e., MacBook) on the network. The communicated task is to compute the fib(x), where x is a large integer value. As shown in the figure, the total CPU time for computing fib(x) is decreased by a factor of 5 after delegating the task of computing the Fibonacci of 50 to a more powerful server.

## 8. Conclusions

In the paper, we presented Secure-Actor-based Model for IoT Communication (SecIoTComm), a model and framework for representing secure IoT communication. SecIoTComm model defines IoT communication compositionally, where we presented a set of composition rules for constructing complex communication by composing simpler ones. We also presented the operational semantics for SecIoTComm, which are defined by a transition relation on SecIoTComm configuration. We also presented the design of a framework implementing the SecIoTComm model, which can be used to easily create complex secure IoT communication. Finally, we conducted several sets of experiments to evaluate the performance and scalability of the framework.

Work is ongoing in multiple directions. First, we would like to study important properties of secure IoT communication, focusing on equivalence. Second, we are looking at building programming constructs implemented in a platform to support the task delegation algorithm without manually setting up the communication infrastructure.

## Figures and Tables

**Figure 1 sensors-22-07313-f001:**
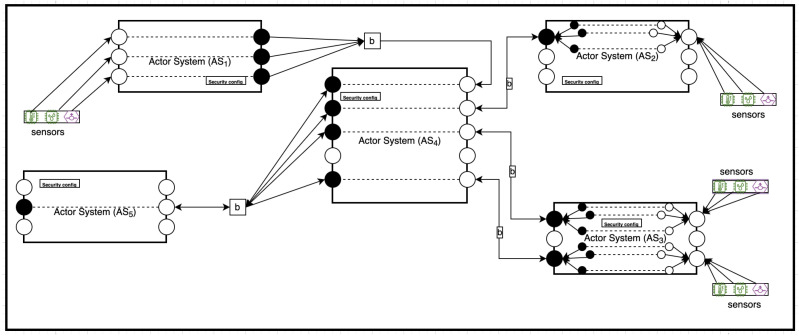
Secure Communication between IoT devices.

**Figure 2 sensors-22-07313-f002:**
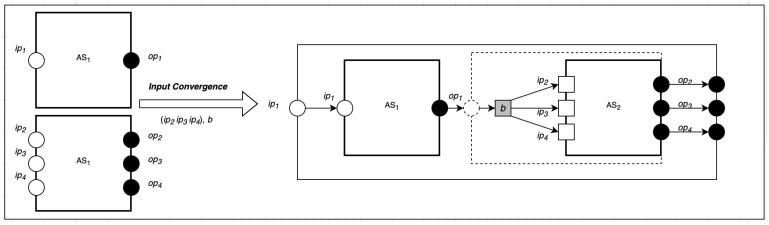
Input Convergence Composition.

**Figure 3 sensors-22-07313-f003:**
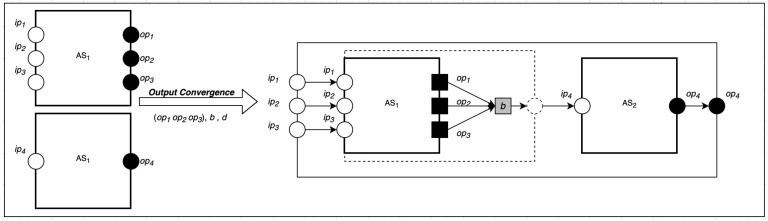
Output Convergence Composition.

**Figure 4 sensors-22-07313-f004:**
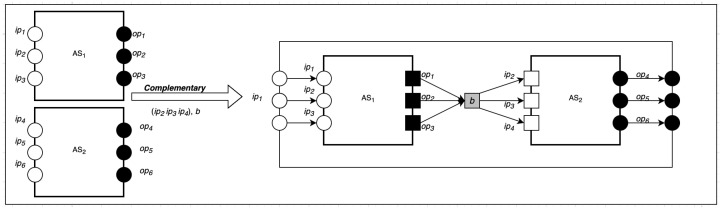
Complementary Composition.

**Figure 5 sensors-22-07313-f005:**
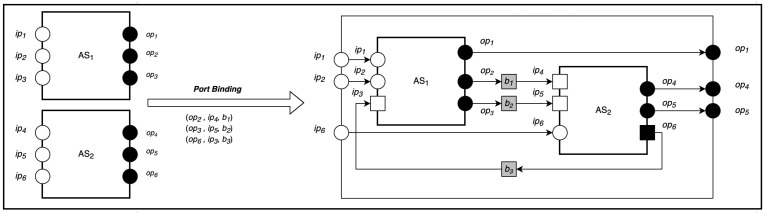
Port-Binding Composition.

**Figure 6 sensors-22-07313-f006:**
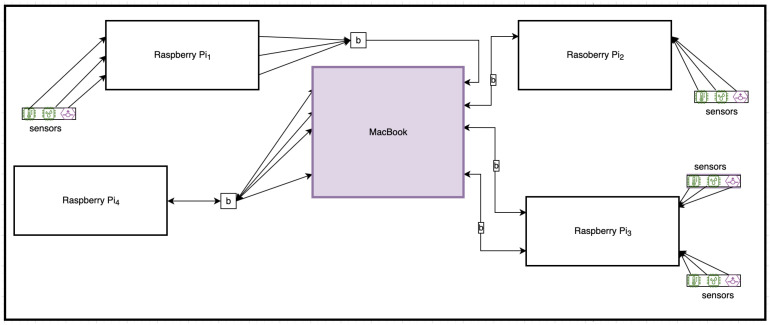
SecIoTComm Implementation Framework.

**Figure 7 sensors-22-07313-f007:**
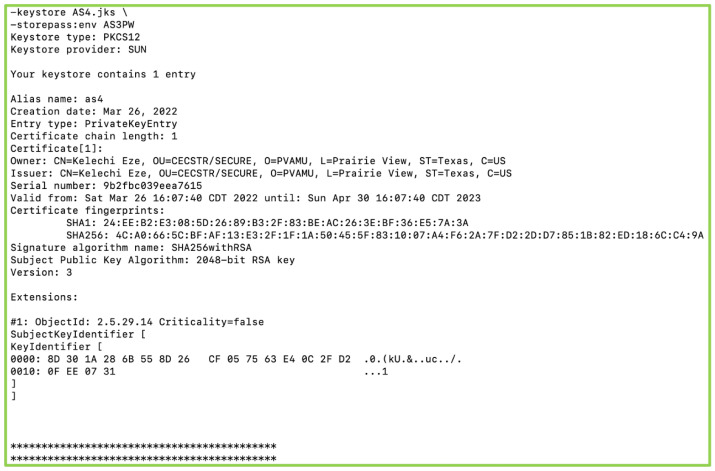
Private Keys.

**Figure 8 sensors-22-07313-f008:**
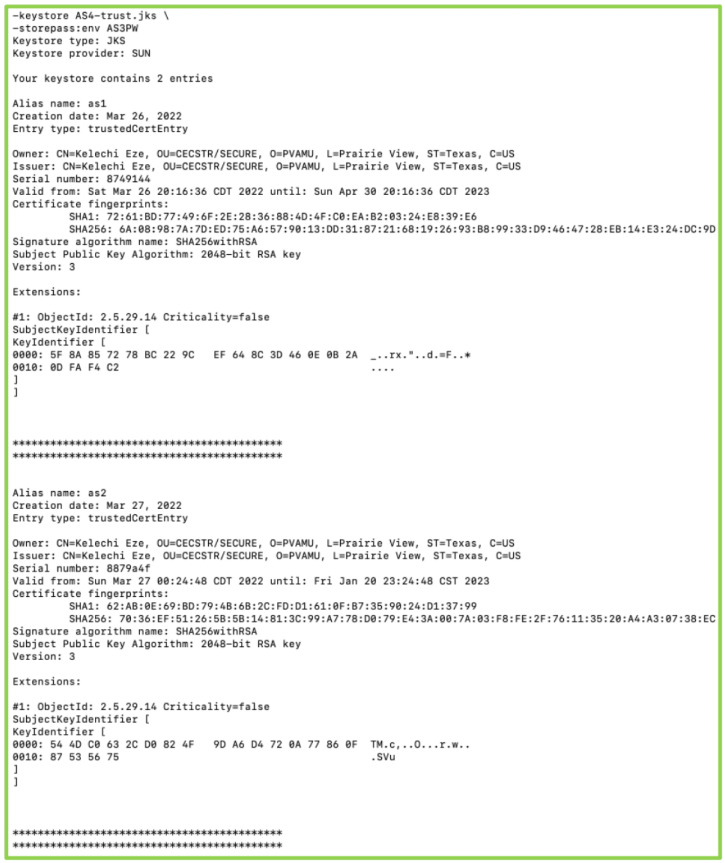
Public keys.

**Figure 9 sensors-22-07313-f009:**
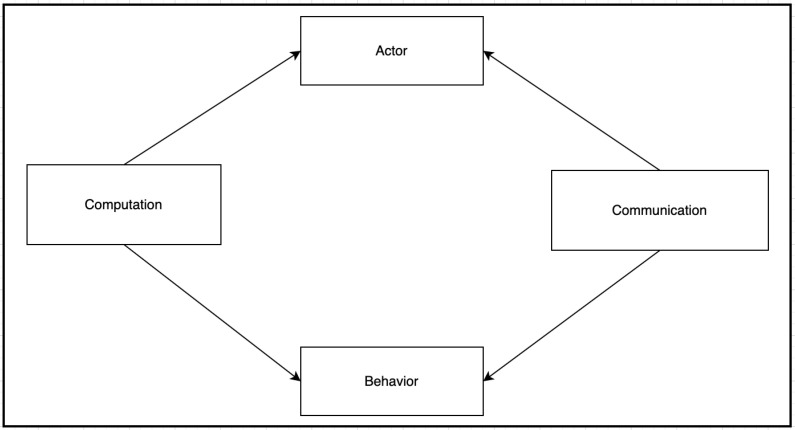
Class Diagram.

**Figure 10 sensors-22-07313-f010:**
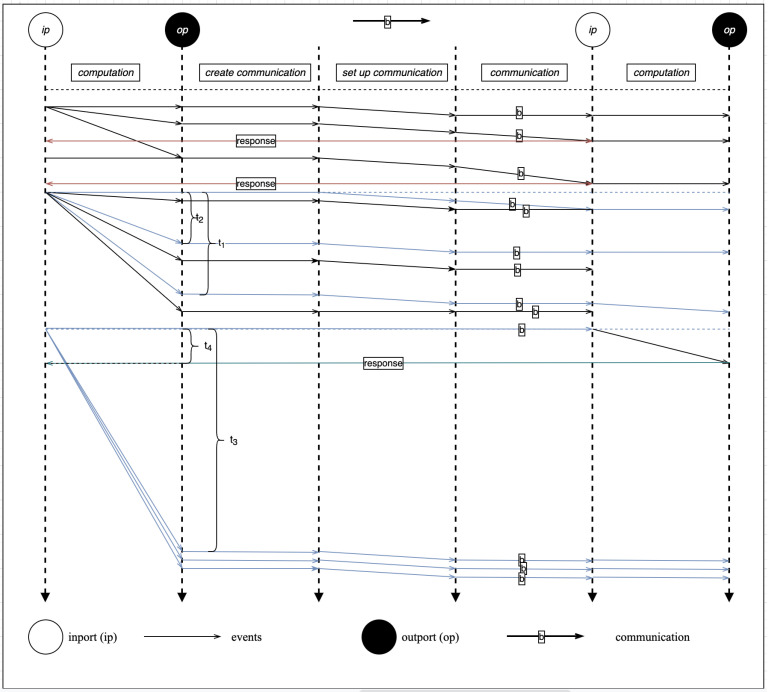
Runtime Event Diagram.

**Figure 11 sensors-22-07313-f011:**
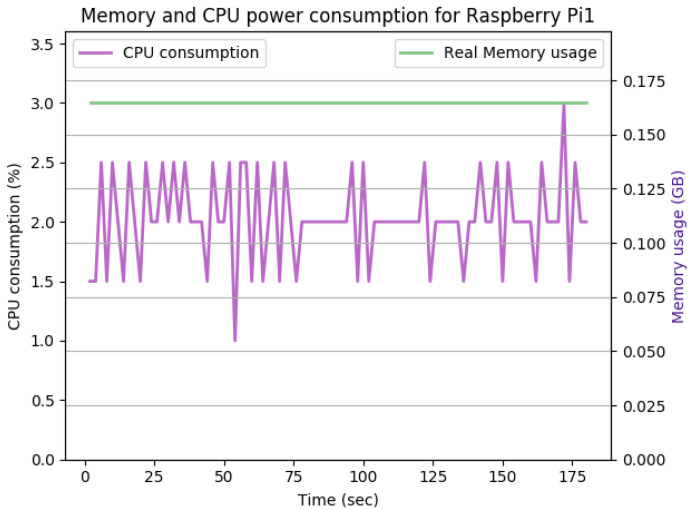
The CPU and memory utilization of the Raspberry Pi device in the idle JVM state.

**Figure 12 sensors-22-07313-f012:**
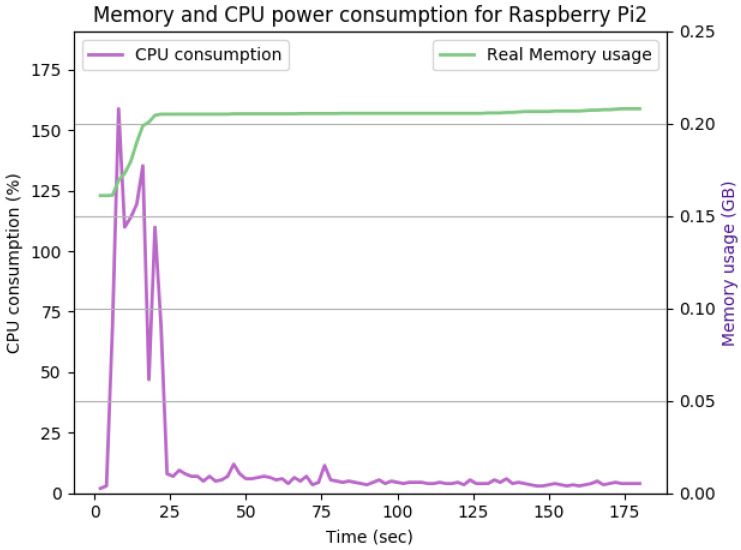
The CPU and memory utilization of the Raspberry Pi device using $TLS\_DHE_{R}SA\_WITH_{A}ES\_128\_GCM\_SHA256$, keysize = 512.

**Figure 13 sensors-22-07313-f013:**
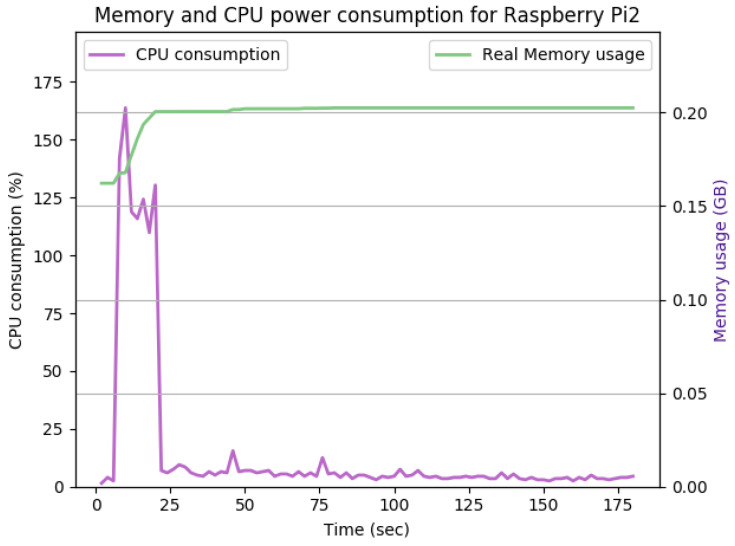
The CPU and memory utilization of the Raspberry Pi device using TLS_DHE_RSA_WITH_AES_128_GCM_SHA256, keysize = 1024.

**Figure 14 sensors-22-07313-f014:**
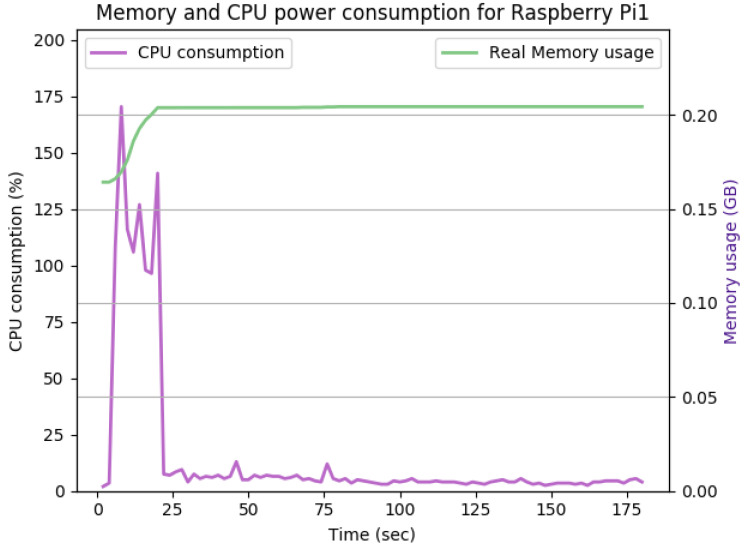
The CPU and memory utilization of the Raspberry Pi device using TLS_DHE_RSA_WITH_AES_128_GCM_SHA256, keysize = 2048.

**Figure 15 sensors-22-07313-f015:**
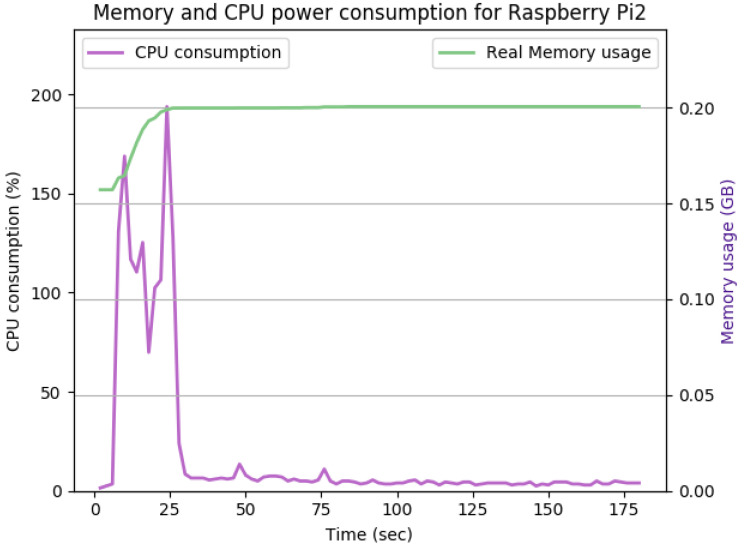
The CPU and memory utilization of the Raspberry Pi device using TLS_DHE_RSA_WITH_AES_128_GCM_SHA256, keysize = 4096.

**Figure 16 sensors-22-07313-f016:**
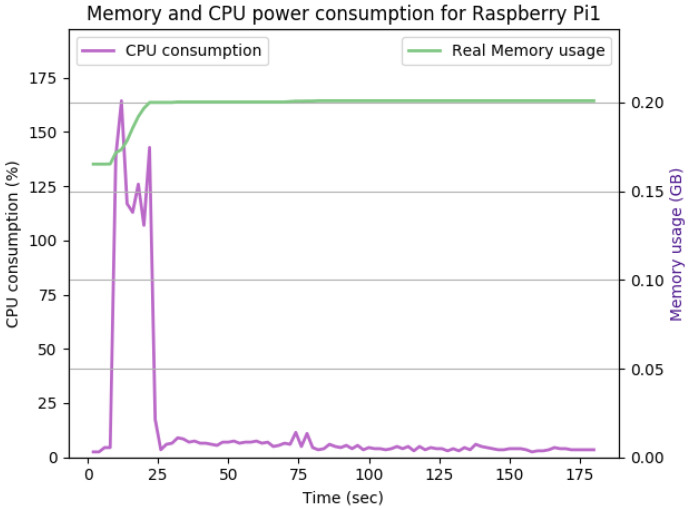
The CPU and memory utilization of the Raspberry Pi device using TLS_ECDHE_RSA_WITH_AES_128_GCM_SHA256, keysize = 2048.

**Figure 17 sensors-22-07313-f017:**
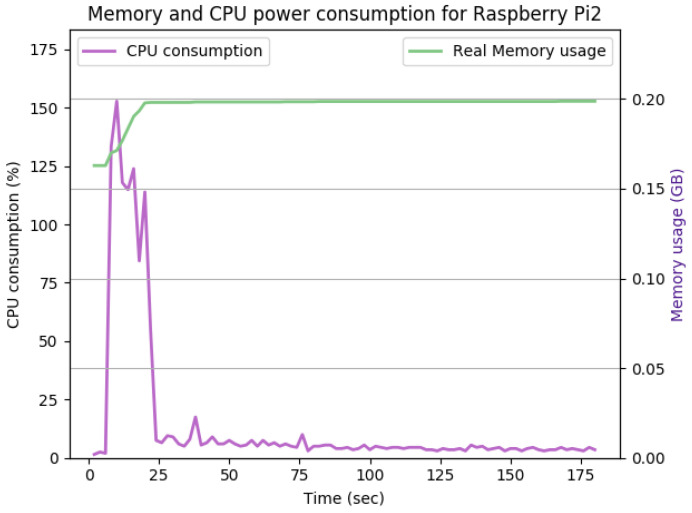
The CPU and memory utilization of the Raspberry Pi device using TLS_ECDHE_RSA_WITH_AES_128_GCM_SHA256, keysize = 512.

**Figure 18 sensors-22-07313-f018:**
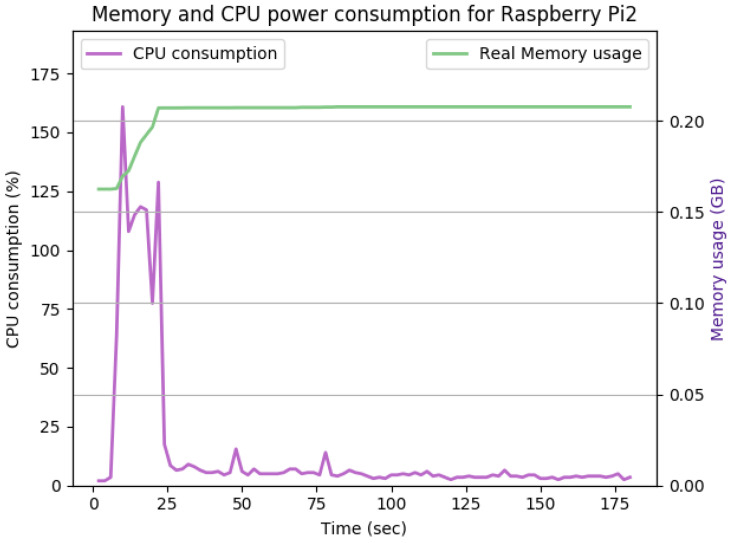
The CPU and memory utilization of the Raspberry Pi device using TLS_ECDHE_RSA_WITH_AES_128_GCM_SHA256, keysize = 1024.

**Figure 19 sensors-22-07313-f019:**
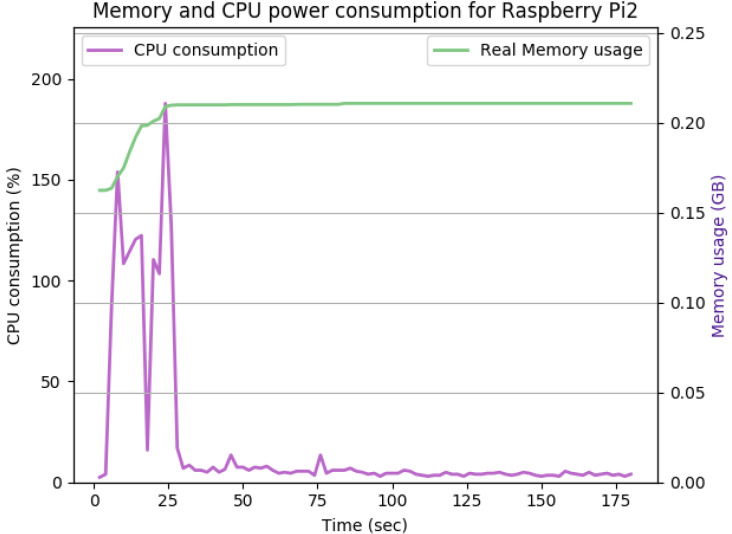
The CPU and memory utilization of the Raspberry Pi device using TLS_ECDHE_RSA_WITH_AES_128_GCM_SHA256, keysize = 4096.

**Figure 20 sensors-22-07313-f020:**
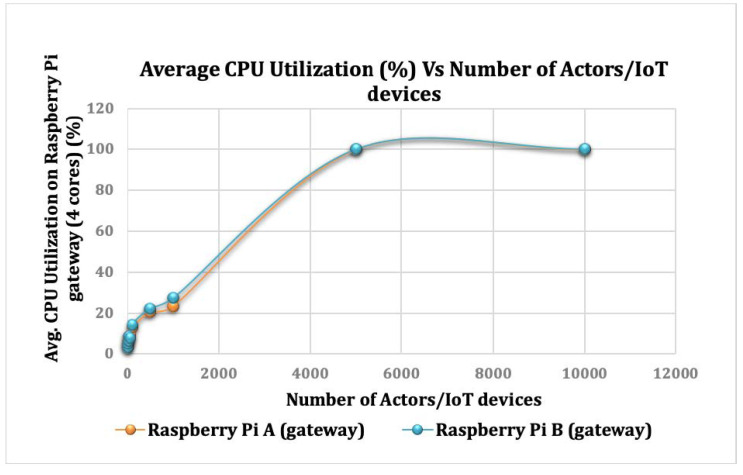
The Effect of Changing the Number of Actors on The CPU utilization of the Raspberry Pi devices.

**Figure 21 sensors-22-07313-f021:**
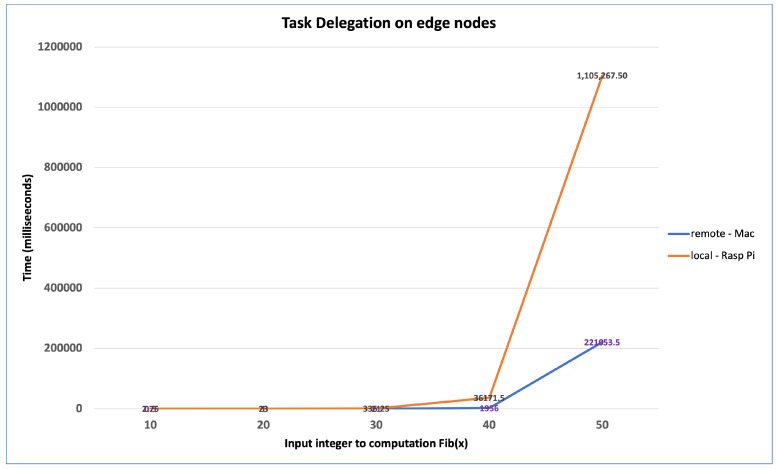
Computational Task Delegation.

**Table 1 sensors-22-07313-t001:** The Mathematical Symbols used in SecIoTComm.

Symbol	Definition
α	A map between actors’ addresses to their behaviors
*A*	A finite set of computational actors
*C*	A finite set of communication actors
*m*	An actor message
*M*	A set of actor messages
*b*	An actor’s behavior
app(b,m)	An application of an actor message *m* to a behavior *b*
conf	A security configuration
(I:T:O)	A computational sequence of inputs, transitions, and outputs
μ	Transient messages
*k*	SecIoTComm configuration which represents a secure IoT communication
op	A set of output actors for sending messages
ip	A set of input actors for receiving messages
χ	A set of external actors
FV	A free variable
R[r]	A reduction context for reducing the expression *r*
$\overset{n}{\underset{i=1}{⋃}}$	A composition of *n* communication configurations

**Table 2 sensors-22-07313-t002:** The Performance of the TLS_DHE_RSA_WITH_AES_128_GCM_SHA256 Experiments.

Key Size	CPU Peak	Avg. CPU	Handshake Duration	Memory Overhead
512	160%	5%	23 s	45 Mb
1024	165%	5%	23 s	35 Mb
2048	170%	5%	23 s	50 Mb
4096	196%	5%	30 s	40 Mb

**Table 3 sensors-22-07313-t003:** The Performance of the TLS_ECDHE_RSA_WITH_AES_128_GCM_SHA256 Experiments.

Key Size	CPU Peak	Avg. CPU	Handshake Duration	Memory Overhead
512	155%	5%	23 s	35 Mb
1024	160%	5%	23 s	45 Mb
2048	164%	5%	23 s	30 Mb
4096	183%	5%	25 s	50 Mb

## Data Availability

The data and source code are available upon request.

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
