# Peer review of "SecIoTComm: An Actor-Based Model and Framework for Secure IoT Communication"

_sensors, 2022, doi:10.3390/s22197313_

Round 1

Reviewer 1 Report

- The work is well organized;

- Theoretical research is combined with practical research and testing of the proposed solution;

- In the paper, the authors present the Secure Actor-based Model for IoT Communication (SecIoT-Comm), a model and framework for representing secure IoT communication.

Recommendations for improvement:

- adding a table highlighting the differences between the proposed solution and the solutions on the market according to the current state of knowledge (to better see the necessity of the study, as well as the contribution to the development of knowledge in the field);

- completion with current bibliographic resources and with references in the text that highlight the contribution of the proposed solution to the current state of knowledge;

- adding in the conclusions section the future directions of research, as well as the limits of the study;

- all the figures in the article will be referred to (see figure 5)

- rereading the work and correcting editing errors.

Author Response

Reviewer#1, Concern # 1: Adding a table highlighting the differences between the proposed solution and the solutions on the market according to the current state of knowledge (to better see the necessity of the study, as well as the contribution to the development of knowledge in the field)

Author response: Thanks for the comment. We added several recent related works of literature related to Section II with a summary and discussion. We also highlighted the novelty of our work compared to existing work in the area of secure IoT communication.

Reviewer#1, Concern # 2: Completion with current bibliographic resources and with references in the text that highlight the contribution of the proposed solution to the current state of knowledge.

Author response: Thanks for the comment. We added several recent related works of literature related to Section II with a summary and discussion such as Ref #21 and 22. We also added a new paragraph by the end of Section II to compare our work to existing work.

Reviewer#1, Concern # 3: Adding in the conclusions section the future directions of research, as well as the limits of the study.

Author response: Thanks for the comment. We added a new paragraph in the Conclusion section to present our future directions of this work and the limits of this study.

Reviewer#1, Concern # 4: all the figures in the article will be referred to (see figure 5)

Author response: Thanks for the comment. We fixed this issue throughout the whole manuscript.

Reviewer#1, Concern # 5: Rereading the work and correcting editing errors.

Author response: Thanks for the comment. We updated the manuscript by fixing the editing errors and drastically enhancing the general presentation of the paper, including graphics, discussion, and exposition.

Author Response

Concern # 1: The background in abstract should be briefly introduced

Author response: Thanks for the comment. We updated the manuscript by adding several paragraphs and sentences distributed across the manuscript to clarify this point. Also, we highlighted the main contributions of this work in the introduction section clearly. The reference list has been updated at the end of this manuscript.

Concern # 2: Please check the whole paper to avoid writing and grammar errors. For instance, 1) “we are present Secure Actor-based Model”; 2) “SecIoTComm aims to understand” should be revised as “SecIoTComm aims to contain and represent”.

Author response: Thanks for the comment. We updated the manuscript by fixing the editing errors and drastically enhancing the general presentation of the paper, including graphics, discussion, and exposition.

Concern # 3: It is suggested to unify the English tence in abstract, such as “we present an IoT framework” and “we evaluated the developed framework”.

Author response: Thanks for the comment. We fixed these types of errors throughout the whole manuscripts.

Concern # 4: It is suggested to introduce the following recent works in advanced wireless communication technique [R1], secure communication [R2]-[R3] fields to highlight

the state-of-the-art of this paper.

Author response: Thanks for the comment. We added several recent related works of literature related to Section II with a summary and discussion such as Ref #21 and 22. We also added a new paragraph by the end of Section II to compare our work to existing work.

Concern # 5: Descriptions and notations should be added into figure 1 to make it more clear

Author response: Thanks for the comment. The description and notations in Figure are described and explained in detail in the text where we refer to Figure 1.

Concern # 6: There are too many mathematical symbols without definition, it is suggested to add a Table to define them.

Author response: Thanks for the comment. We added Table 1 that summarizes the mathematical symbols used in SecIoTComm.

Concern # 7: The motivations and contributions should be further clarified to emphasize the novelty of this paper.

Author response: Thanks for the comment. We updated the manuscript by adding several paragraphs and sentences distributed across the manuscript to clarify this point. For instance, we added a new paragraph at the end of the Introduction section to highlight the main contributions of this paper: “The contributions of this paper are fourfold. First, we present the syntax and operational semantics of SecIoTComm. The syntax defines the SecIoTComm configuration, and the semantic represents the meaning of secure IoT operations, including communication, computations, and security operations. Second, we show how complex communications can be built by composing simpler ones, and give examples to demonstrate the concept of communication composition. SecIoTComm can be used to understand and represent the secure communication and coordination requirements of a wider class of sensor-based systems. Also, we present the syntax and operational semantics of these composition rules with examples. Third, we present an actor-based framework implementing the SecIoTComm model using Scala/Akka programming language. Finally, we evaluated the developed framework using various experimental experiments.”

Reviewer 3 Report

The manuscript proposes Secure Actor-based Model for IoT Communication (SecIoTComm), a model for representing secure IoT communication. SecIoTComm aims to understand secure IoT communications’ properties and design and implement novel mechanisms to improve their programmability and performance. Nevertheless, the following comments require to consider in this round.

(1) In line 51, authors are recommended to replace bold font with uppercase letters.

(2) In line 61, authors are recommended to rewrite and highlight contributions by drawing separated points.

(3) Authors are recommended to follow the following presentation to cite references.  As an example in line 85 ((Ahmed et al. [13,14] interpreted and implemented …))

 (4) In the related work section, authors are recommended to add a new paragraph at the end of the section to compare their work with others.

 (5) Rewrite the section of 3. Secure Actor-based Model for IoT Communication by drawing points to each component in the system model.

 (6) Please add figures and flowcharts to present your proposed work in the manuscript.

(7)A comparison between your work and recent work is missing.

(8) authors are recommended to show improvement outperforms among works in abstract

Author Response

Concern # 1: In line 51, authors are recommended to replace bold font with uppercase letters.

Author response: Thanks for the comment. Since the abbreviation’s letters are not all capital letters, we think it would be better to refer to SecIoTComm as a Secure Actor-based Model for IoT Communication.

 Concern # 2: In line 61, authors are recommended to rewrite and highlight contributions by drawing separated points.

Author response: Thanks for the comment. We rewrote this paragraph to highlight the main contributions of the work as four points.

Concern # 3: Authors are recommended to follow the following presentation to cite references.  As an example, in line 85 (Ahmed et al. [13,14] interpreted and implemented …)

Author response: Thanks for the comment. We think that changing the presentation of citations will make it easier for readers to follow up the points behind each citation.

Concern # 4: In the related work section, authors are recommended to add a new paragraph at the end of the section to compare their work with others.

Author response: Thanks for the comment. We added several recent related works of literature related to Section II with a summary and discussion such as Ref #21 and 22. We also added a new paragraph by the end of Section II to compare our work to existing work.

Concern # 5: Rewrite the section of 3. Secure Actor-based Model for IoT Communication by drawing points to each component in the system model.

Author response: Thanks for the comment. We modified Section 3 to address this point.

Concern # 6: Please add figures and flowcharts to present your proposed work in the manuscript

Author response: Thanks for the comment. Thanks for the comment. We updated the manuscript by fixing the editing errors and drastically enhancing the general presentation of the paper, including graphics, discussion, and exposition.

Concern # 7: A comparison between your work and recent work is missing.

Author response: Thanks for the comment. We added a new paragraph by the end of Section II to compare our work to existing work.

Concern # 8: Authors are recommended to show improvement outperforms among works in abstract.

Author response: Thanks for the comment. We showed how our work outperforms the existing work in the literature as described in Section II.

Round 2

Reviewer 2 Report

The authors have basically addressed my concerns, no further comments.

Reviewer 3 Report

The authors paid extra attention to address my concerns. Thus, I am decided to accept this paper in the present form.